# Current Status of Precision Medicine in Colorectal Cancer in Japan

**DOI:** 10.3390/ijms26115029

**Published:** 2025-05-23

**Authors:** Yoshiki Kojitani, Masayuki Takeda

**Affiliations:** Department of Cancer Genomics and Medical Oncology, Nara Medical University, 840 Shijo-Cho, Kashihara 634-8521, Nara, Japan; y.kojitani@naramed-u.ac.jp

**Keywords:** precision medicine, human epidermal growth factor receptor 2, molecular targeted therapy, colorectal cancer, Japan

## Abstract

Colorectal cancer (CRC) remains a major health burden in Japan, with precision medicine playing an increasingly critical role in treatment optimization. Key biomarkers, including *RAS*, *BRAF*, microsatellite instability/mismatch repair, and human epidermal growth factor receptor 2, can be used as a guide for molecularly targeted therapies and immunotherapy. Advances in molecular diagnostics, including comprehensive genomic profiling, have enabled more precise treatment selection such as *RET* and *NTRK* fusions. Nationwide initiatives, such as c-CAT and SCRUM-Japan, can leverage real-world data to refine clinical strategies. Recent developments in circulating tumor DNA analysis have led to novel approaches for minimal residual disease monitoring, as demonstrated by the CIRCULATE-Japan GALAXY study. However, certain challenges persist, including the time required for genetic testing, the limited availability of targeted therapies, and disparities in access to molecular tumor boards. This review summarizes the current landscape of precision medicine in CRC in Japan, emphasizing key biomarkers, genetic testing strategies, targeted therapies, and emerging technologies. Future research should focus on expanding clinical trial access, accelerating drug approvals, and integrating real-world data into clinical practice to further advance precision medicine.

## 1. Introduction

Colorectal cancer (CRC) is one of the most prevalent and fatal malignancies occurring in Japan as well as worldwide, highlighting the need for improvements in its early diagnosis and treatment [1]. Beyond conventional chemotherapy, recent advancements in molecular targeted therapy and immunotherapy have highlighted the significance of precision medicine. In CRC, biomarkers such as *RAS* (*KRAS*, *NRAS*), *BRAF*^V600E^, microsatellite instability (MSI)/mismatch repair, and human epidermal growth factor receptor 2 (*HER2*) can significantly influence treatment selection, enabling the development of tailored therapeutic strategies based on specific genetic mutations.

Over the past few decades, CRC treatment has evolved markedly. Traditional chemotherapy based on 5-fluorouracil has improved after the introduction of combination regimens such as FOLFOX and FOLFIRI, leading to better treatment outcomes [2,3]. Furthermore, the advent of molecular targeted agents, including epidermal growth factor receptor (EGFR) inhibitors (cetuximab and panitumumab) [4,5,6,7] and vascular endothelial growth factor (VEGF) inhibitors (bevacizumab, ramucirumab, and aflibercept) [8,9,10,11,12], has enabled more effective treatment. However, the efficacy of these molecular targeted therapies varies and depends on the molecular profile of the tumor. For instance, anti-EGFR antibody therapy has proved ineffective in patients with *RAS* mutations, making molecular diagnostics indispensable for appropriate treatment selection [13]. In addition, *BRAF*^V600E^ mutations are associated with extremely poor prognosis, and conventional chemotherapy alone has limited efficacy, necessitating treatment strategies that include *BRAF*^V600E^ and *MEK* inhibitors [14]. Japan has made notable advances in standardizing genetic testing and expanding insurance coverage, facilitating precision medicine implementation.

Notably, the Center for Cancer Genomics and Advanced Therapeutics (c-CAT) and the SCRUM-Japan consortium serve as centralized platforms for real-world data collection and trial enrollment. The Japanese government also established a designated network of Cancer Genomic Medicine Hospitals to facilitate CGP interpretation and therapy matching. These developments contrast with genomic programs in the United States and Europe, where broader insurance coverage, earlier access to CGP, and a higher rate of biomarker-matched therapy adoption have been observed.

Despite progress, multiple challenges persist in Japan, including delays in test turnaround times, limited access to off-label or investigational therapies, and variation in molecular tumor board (MTB) operations. Moreover, key technologies such as ctDNA-based monitoring and DNA methylation assays remain outside the scope of reimbursement, limiting their clinical utility.

In this review, we critically examine the current landscape of precision medicine in CRC in Japan. We provide an updated synthesis of key biomarkers, diagnostic modalities, therapeutic strategies, and national initiatives. We also highlight emerging technologies, discuss implementation barriers, and compare Japan’s genomic medicine framework with global benchmarks. Finally, we offer recommendations to enhance the equitable and effective integration of precision medicine into routine CRC care in Japan.

## 2. Diagnostics Covered by Japanese Medical Insurance

### 2.1. Biomarker Testing

Recent years have witnessed notable advances in personalized treatment strategies for CRC that emphasize the importance of biomarker-based therapy selection. In Japan, biomarker tests for *RAS* and *BRAF* mutations, *HER2* amplification/overexpression, MSI/MMR status, and CGP are covered by insurance and widely utilized in clinical practice.

*RAS* mutation testing is essential for determining the eligibility of the anti-EGFR antibody. In Japan, the MEBGEN RASKET™-B Kit is an approved diagnostic tool that detects mutations in exons 2, 3, and 4 of *KRAS* and *NRAS* [15,16]. Although this test is primarily performed using tumor tissue, circulating tumor DNA (ctDNA) testing utilizing the OncoBEAM™ RAS CRC Kit becomes an option when tissue samples are unavailable [17].

*BRAF* mutation testing focuses on identifying *BRAF*^V600E^ mutations, a predictor of poor prognosis and a key biomarker for *BRAF* and *MEK* inhibitor therapies. The MEBGEN RASKET™-B Kit and Therascreen^®^ BRAF V600E RGQ PCR Kit are approved for detecting *BRAF*^V600E^ mutations [15,16]. The guidelines recommend *BRAF* mutation testing before initiating first-line therapy in unresectable advanced or recurrent CRC.

*HER2* testing is necessary for selecting patients eligible for *HER2*-targeted therapies, such as *HER2* amplification or overexpression. According to the latest guideline, *HER2* testing should be conducted before administering anti-*HER2* therapies. The evaluation involves immunohistochemistry (IHC) followed by fluorescence in situ hybridization (FISH) for equivocal (IHC 2+) cases [18].

MSI/MMR testing is essential for determining eligibility for immune checkpoint inhibitors (ICIs) for advanced and recurrent CRC [19,20,21]. The latest guideline strongly recommends MSI/MMR testing before first-line treatment initiation in patients with unresectable advanced or recurrent disease. MSI testing is performed using polymerase chain reaction (PCR), and MMR testing is performed using IHC to evaluate the status of key MMR proteins (*MLH1*, *MSH2*, *MSH6*, and *PMS2*). In addition, MSI/MMR testing plays a crucial role in Lynch syndrome screening [22].

### 2.2. CGP

CGP is an advanced NGS-based technique that allows for the simultaneous detection of multiple genetic alterations, including mutations, gene fusions, amplifications, MSI, and tumor mutation burden (TMB) [23]. In CRC, CGP plays a crucial role in evaluating key biomarkers such as *RAS*, *BRAF*, *HER2*, MSI, TMB, *RET* fusions, and *NTRK* fusions, which are essential for selecting molecularly targeted therapies and ICIs.

In Japan, CGP testing has been covered by national health insurance since June 2019 for patients with solid tumors who have completed standard treatment or those for whom no standard treatment is available. For rare cancers, CGP testing is covered even at the initial treatment stage, whereas, for other cancers, insurance coverage is limited to cases in which standard treatment has been completed or is planned to be completed. In addition, insurance-covered CGP testing can be performed only once in a lifetime per patient. Japan operates under a universal healthcare system, where patients typically bear only 10–30% of medical costs depending on their age and income level. To further mitigate financial burden, the High-Cost Medical Expense Benefit System caps monthly out-of-pocket payments. These frameworks support broader access to CGP and biomarker testing, even though regional disparities in implementation and institutional experience may persist. Currently, insurance-covered CGP tests include OncoGuide™ NCC Oncopanel System and FoundationOne^®^ CDx as tissue-based tests, while FoundationOne^®^ Liquid CDx and Guardant360^®^ CDx are available as blood-based tests. GenMineTOP^®^ requires tissue and blood samples for analysis (Table 1). Tissue-based tests analyze formalin-fixed paraffin-embedded samples, while blood-based tests detect genetic alterations through ctDNA analysis. The number of genes analyzed and the detection capabilities of each test vary, with details provided in Table 1.

The introduction of CGP has substantially advanced precision medicine in Japan by enabling the identification of actionable mutations across multiple genes. However, several implementation challenges remain. Compared with single-gene assays, CGP requires a longer turnaround time, which can delay clinical decision making. Furthermore, despite the detection of potentially actionable alterations in nearly 40% of patients, only about 8% ultimately receive genomically matched therapies [24]. This gap is attributable to multiple structural and regulatory barriers, including the limited availability of approved targeted agents, difficulties in accessing clinical trials due to eligibility criteria and site limitations, and constrained pathways for off-label drug use. While Japan offers mechanisms such as advanced medical care, clinical trials, and patient-requested therapy, each is associated with procedural or infrastructural hurdles that limit broad applicability.

In response to these limitations, Japan has established molecular tumor boards (MTBs) at designated cancer genomic medicine hospitals. These multidisciplinary teams are intended to aid clinicians in interpreting CGP results and recommending appropriate treatment strategies. However, variation in the structure, expertise, and operational protocols of MTBs across institutions has led to inconsistent clinical application.

In contrast to Japan, where CGP is largely restricted to patients who have completed standard treatments, the United States allows broader access. Under the National Coverage Determination policy, Medicare covers CGP for patients with advanced or recurrent cancers without restrictions on timing [25]. Moreover, clinical studies such as Know Your Tumor and I-PREDICT have shown improved survival with genomically matched therapies, highlighting the clinical value of timely CGP implementation [26,27].

To improve access to investigational therapies, national initiatives such as the MASTER KEY project have been launched, providing a centralized registry to connect CGP-tested patients with appropriate clinical trials [28]. Moving forward, expanding the list of reimbursed targeted agents, promoting adaptive trial designs such as basket and umbrella studies, implementing decentralized clinical trials, and creating more flexible pathways for accessing unapproved drugs will be essential to close the gap between genomic data and clinical benefit.

## 3. Key Biomarkers and Their Clinical Implications

### 3.1. RAS (KRAS, NRAS) Mutations

*RAS* (*KRAS* and *NRAS*) encode low-molecular-weight GTP-binding proteins located downstream of the EGFR signaling pathway, which play crucial roles in cell proliferation, survival, and differentiation. *RAS* (*KRAS* and *NRAS*) mutations critically influence the efficacy of anti-EGFR antibody therapy. Initially, *KRAS* exon 2 mutations were shown to confer resistance, and subsequent analyses expanded this exclusion to mutations in *KRAS* exons 3 and 4 and *NRAS* [5,7,29,30,31]. As such, current guidelines recommend anti-EGFR therapy only for *RAS* wild-type CRC. Although some reports have suggested that patients with *KRAS* G13D mutations can benefit from cetuximab, results from large-scale clinical trials have been inconsistent, and established evidence supporting the use of anti-EGFR inhibitors for these cases remains lacking [32].

More recently, several clinically important developments have emerged. Recent studies have reported that, even in *RAS* wild-type CRC, differences in DNA methylation status can affect the efficacy of anti-EGFR antibody therapy. Highly methylated colorectal cancer (HMCC) cases exhibit limited response to anti-EGFR therapy, indicating treatment resistance similar to that observed in *RAS*-mutant tumors. This is further discussed in the DNA methylation section [33].

Among *KRAS* mutations, *KRAS* G12C has recently gained attention as a potential therapeutic target. *KRAS* G12C mutations occur in approximately 3% of CRCs [34]. *KRAS* G12C-specific inhibitors, such as sotorasib and adagrasib, are designed to selectively bind to the GDP-bound form of *KRAS* G12C, locking it in an inactive state and suppressing aberrant signaling. However, in Phase I/II trials evaluating these inhibitors in patients with advanced or recurrent *KRAS* G12C-mutant CRC, monotherapy with sotorasib had only limited efficacy [35]. Conversely, combination therapy with anti-EGFR inhibitors showed improved response rates. This enhanced effect is believed to result from the reliance of *KRAS* G12C-mutant CRC tumors on EGFR signaling, suggesting that the dual inhibition of both pathways can provide a more effective therapeutic approach. Similarly, clinical trials on adagrasib for *KRAS* G12C-mutant CRC have demonstrated promising results when combined with EGFR inhibitors [36,37].

These results have led to the FDA approval of adagrasib plus cetuximab and sotorasib plus panitumumab for *KRAS* G12C-mutant CRC, markedly expanding the treatment options for these patients [36,37]. However, these therapies remain unapproved in Japan. Meanwhile, new drugs targeting other *KRAS* mutations, such as *KRAS* G12D and *KRAS* G12V, are currently under development and show promising potential for future clinical applications. Furthermore, efforts are underway to develop pan-*RAS* inhibitors to address an unmet need in precision medicine [38].

In Japan, *RAS* mutation testing has been covered by insurance since April 2015, restricting the use of anti-EGFR inhibitors to *RAS* wild-type cases (*KRAS*/*NRAS* exon 2, 3, and 4 mutation-negative). The JSMO guidelines recommend *RAS* mutation testing before initiating first-line treatment for advanced and recurrent CRC to determine eligibility for anti-EGFR antibodies. The MEBGEN RASKET™-B Kit is commonly used for this purpose, employing PCR-rSSO (PCR-reverse sequence-specific oligonucleotide) technology to detect *KRAS* and *NRAS* mutations in exons 2, 3, and 4 using tumor tissue specimens [15,16]. In cases where the tumor tissue is unavailable or unsuitable for genetic testing, liquid biopsy-based *RAS* mutation testing serves as an alternative. The OncoBEAM™ RAS CRC Kit, which utilizes BEAMing (Beads, Emulsions, Amplification, and Magnetics) digital PCR technology, allows for the detection of *RAS* mutations in ctDNA from plasma samples [17]. A particularly noteworthy emerging concept is Neo-*RAS* wild-type CRC, which may redefine treatment eligibility for anti-EGFR therapies. This term refers to cases in which a tumor that was initially *RAS*-mutant at diagnosis later becomes *RAS* wild-type during the course of treatment. This phenomenon has been observed through liquid biopsy, where ctDNA analysis reveals the loss of *RAS* mutations following systemic therapy [38]. Patients with Neo-*RAS* wild-type status may experience significant clinical benefits from reintroducing EGFR inhibitors [39,40,41].

### 3.2. BRAF^V600E^ Mutations

*BRAF* encodes a serine/threonine kinase that plays a central role in the *MAPK* signaling pathway, regulating cell proliferation and survival. *BRAF* mutations are detected in approximately 5–10% of CRC cases, with about 90% being the V600E mutation [42,43]. This mutation results in the substitution of valine (Val) with glutamic acid (Glu), leading to a conformational change in the *BRAF* protein that enables constitutive activation of the *MAPK* pathway independent of *RAS* regulation. Consequently, *BRAF*^V600E^ mutations drive excessive tumor growth and confer a poor prognosis in CRC.

Clinically, *BRAF*^V600E^-mutant CRC is more frequently observed in right-sided tumors (cecum and ascending colon), older patients, and females. In addition, *BRAF*^V600E^ mutations are often associated with MSI-high (MSI-H), a characteristic that predicts favorable responses to ICIs [42,43]. In contrast, microsatellite stable (MSS) *BRAF*^V600E^-mutant CRC exhibits poor responses to conventional chemotherapy, with standard regimens such as FOLFOX and FOLFIRI plus anti-EGFR antibody showing limited efficacy [14,44].

For CRC harboring *BRAF*^V600E^ mutations, targeted therapy combining a *BRAF* inhibitor and an anti-EGFR antibody has become a standard treatment approach. The pivotal Phase III BEACON CRC trial established the clinical basis for this strategy [45]. The study evaluated two regimens: a doublet combination of encorafenib plus cetuximab and a triplet regimen adding binimetinib. Both regimens significantly improved OS and the objective response rate (ORR) compared to standard chemotherapy. Specifically, the median OS was 9.0 months for the triplet, 8.4 months for the doublet, and 5.4 months for the control, with an ORR of 26%, 20%, and 2%, respectively. However, the study was not statistically powered to compare the doublet and triplet arms directly, and the hazard ratio for OS (HR = 0.79, 95% CI: 0.59–1.06) between the triplet and doublet regimens did not reach significance. Moreover, the triplet was associated with a higher rate of adverse events (Grade ≥ 3: 58% vs. 50%).

As a result, most international guidelines, including the NCCN, recommend the doublet regimen as the standard due to its comparable efficacy and more favorable safety profile. In contrast, the triplet combination has been approved in Japan as a second-line treatment for *BRAF*^V600E^ mutant CRC. This may reflect regulatory emphasis on response rates, potential benefits of *MEK* inhibition in select subgroups, and early submission timing. Nonetheless, the added clinical benefit of *MEK* inhibition in this setting remains to be fully clarified.

The BREAKWATER trial evaluates the use of encorafenib + cetuximab in combination with chemotherapy as a first-line treatment for *BRAF*^V600E^-mutant CRC. The preliminary results have been promising, suggesting that adding encorafenib and cetuximab to standard chemotherapy regimens (FOLFOX or FOLFIRI) can improve response rates and survival outcomes [46]. Based on these findings, encorafenib + cetuximab in combination with chemotherapy may become an option for first-line treatment in the near future.

Unlike *BRAF*^V600E^ mutations, *BRAF*^non-V600E^ exhibited distinct biological and clinical characteristics, with a generally better prognosis than *BRAF*^V600E^ mutations. However, an optimal therapeutic strategy for these mutations remains unestablished, highlighting the need for further research.

In Japan, *BRAF* mutation testing is recommended before initiating first-line treatment. PCR-based assays such as the MEBGEN RASKET™-B Kit is widely used and also covered by national insurance [15,16]. For patients with *BRAF*^V600E^ mutations, first-line treatment considerations include FOLFOXIRI + bevacizumab [47], whereas second-line therapy comprises the three-drug regimen (encorafenib + binimetinib + cetuximab) [45], the standard in Japan. In addition, given the results of the BREAKWATER trial [46], chemotherapy + encorafenib + cetuximab may become a first-line treatment option in the future.

### 3.3. MSI/MMR Status

MSI and MMR statuses are key molecular markers involved in CRC development and progression. MSI occurs due to defects in the MMR system, which corrects DNA replication errors. The major genes involved in MMR include *MLH1*, *MSH2*, *MSH6*, and *PMS2*. When these genes are impaired, the MMR system fails, leading to an accumulation of replication errors, referred to as MSI-H [48]. In contrast, tumors with a functional MMR system are classified as MSS. Thus, MSI-H and dMMR significantly overlap. Nearly all MSI-H CRCs are TMB-high (TMB-H). Moreover, a seminal analysis of 6004 CRC cases using comprehensive profiling (FoundationOne^®^ CDx) reported that 99.7% of MSI-H tumors were also TMB-H [49]. Therefore, the Japanese CRC guidelines state to use either MSI testing or MMR-IHC to determine eligibility for PD-1 antibodies.

MSI-H CRC exhibits distinct histopathological characteristics, including a high proportion of poorly differentiated adenocarcinoma, mucinous carcinoma, and signet-ring cell carcinoma [50]. These tumors often demonstrate medullary growth patterns, with characteristic Crohn-like lymphoid reactions and tumor-infiltrating lymphocytes. Due to these immune-related histological features, MSI-H CRC generally has a favorable prognosis and a lower risk of recurrence than MSS CRC. The prevalence of MSI-H CRC varies by population, accounting for approximately 15% of CRC cases in Western countries and 6–7% in Japan [51,52]. In addition, the proportion of MSI-H tumors decreases in advanced stages, with only 1.9–3.7% stage IV cases reported in Japan [53].

The predictive value of the MSI-H status in adjuvant chemotherapy (ACT) has been well established. In stage II MSI-H CRC, as fluoropyrimidine (FP) monotherapy is associated with worse prognosis, observation without adjuvant therapy is recommended [54]. In stage III MSI-H CRC, ACT remains the standard of care; however, a pooled analysis of Western studies revealed that, although FP monotherapy provided no survival benefit in MSI-H CRC, the addition of oxaliplatin improved outcomes [55]. Based on this evidence, the guidelines recommend against FP monotherapy, reinforcing the necessity of MSI/MMR testing before initiating ACT. If MSI/MMR results are unavailable before initiating therapy, oxaliplatin-containing regimens, such as FOLFOX and CAPEOX, should be selected to ensure efficacy in potential MSI-H cases.

In metastatic and recurrent CRC, MSI-H status is a key determinant for selecting ICIs. The KEYNOTE-177 trial showed that pembrolizumab (anti-PD-1 antibody) significantly prolonged PFS compared with chemotherapy in previously untreated MSI-H metastatic CRC, establishing ICIs as a first-line treatment [19]. Moreover, the CheckMate-142 trial confirmed the efficacy of nivolumab monotherapy and nivolumab plus ipilimumab combination therapy in pretreated MSI-H CRC [20]. Furthermore, the CheckMate-8HW trial evaluated the efficacy of nivolumab plus chemotherapy as a first-line treatment, with results anticipated to further refine treatment strategies [56].

Neoadjuvant immunotherapy has also demonstrated promising efficacy in MSI-H tumors, as reported in the NICHE-1 and NICHE-2 trials. These trials evaluated nivolumab plus ipilimumab in untreated, respectable, locally advanced dMMR/MSI-H CRC. In NICHE-1, neoadjuvant nivolumab plus ipilimumab in 32 patients resulted in a pathologic complete response (pCR) rate of 60% and a major pathologic response rate of 100% [57]. The NICHE-2 trial, which included 83 patients, demonstrated even higher efficacy, with 95% of patients achieving major pathologic response and 65% achieving pCR [58]. These findings indicate that preoperative immunotherapy is highly effective in MSI-H/dMMR CRC. However, neoadjuvant immunotherapy, as evaluated in NICHE trials, remains unapproved in Japan.

MSI/MMR testing is crucial not only during treatment selection but also for screening for Lynch syndrome. Approximately 20–30% of dMMR cases are linked to Lynch syndrome, highlighting the importance of MSI/MMR testing in hereditary CRC diagnosis. Owing to the ethical implications associated with hereditary cancer syndromes, clinicians must ensure proper patient counseling and obtain informed consent prior to MSI/MMR testing. Young-onset CRC cases or those with a relevant family history should be actively considered for screening.

In Japan, MSI/MMR testing is recommended post-surgery or before first-line treatment to guide chemotherapy or immunotherapy selection. For stage II/III CRC, MSI status is critical for determining ACT eligibility, and post-surgical MSI/MMR testing is required. If MSI/MMR results are unavailable before initiating ACT, oxaliplatin-containing regimens should be used to ensure efficacy [55]. In advanced and metastatic CRC, MSI/MMR testing is strongly recommended before first-line treatment to determine ICI eligibility. Thus, MSI/MMR testing is indispensable for both ACT selection and immunotherapy decision making, requiring timely implementation.

### 3.4. HER2 Amplification

*HER2* is a receptor-type tyrosine kinase belonging to the EGFR family. It activates the signaling pathways involved in cell proliferation, differentiation, and apoptosis inhibition. While *HER2* gene amplification and protein overexpression have been well established as therapeutic targets in breast and gastric cancers, they have also been identified in a specific subgroup of CRC, making *HER2* a promising treatment target. *HER2*-positive CRC accounts for approximately 3–5% of all CRC cases, with a higher prevalence in *RAS* wild-type patients [59,60].

Two pivotal clinical trials—TRIUMPH and MyPathway—have demonstrated the efficacy of trastuzumab plus pertuzumab in *HER2*-positive metastatic CRC [61,62]. TRIUMPH was conducted in Japan and limited to *RAS* wild-type patients, while MyPathway, a U.S.-based basket trial, included all *RAS* statuses but showed responses mainly in the wild-type subgroup. Both studies reported similar outcomes with overall response rates of ~30%. Table 2 summarizes key differences in study design and outcomes between the two trials. These findings led to the approval of trastuzumab plus pertuzumab in Japan. In addition, pertuzumab/trastuzumab/hyaluronidase (Phesgo^®^), the subcutaneous formulation of trastuzumab plus pertuzumab, has been approved in Japan, providing a more convenient administration option for patients.

Furthermore, the DESTINY-CRC01 trial evaluated the efficacy of the antibody-drug conjugate (ADC) trastuzumab deruxtecan (T-DXd) in *HER2*-positive CRC, demonstrating a promising ORR of 45.3% and a median PFS of 6.9 months [63]. In addition, the FDA has approved T-DXd as a pan-tumor therapy for IHC 3+ tumors [64], which suggests that T-DXd can also become available for IHC 3+ CRC in Japan in the near future. In Japan, *HER2* testing is recommended before initiating first-line treatment. *HER2* positivity is determined by IHC or FISH testing on tumor tissue, with *HER2* positivity defined as IHC 3+ or IHC 2+ with FISH positivity [18]. Given that *HER2*-positive CRC has a poor response to anti-EGFR antibody therapy, combination therapy with antiangiogenic agents may be considered [65]. This is based on the hypothesis that *HER2* overactivation limits the effectiveness of EGFR inhibition by signaling downstream of the receptor. Therefore, *HER2* testing should be performed before starting first-line treatment to guide treatment selection appropriately.

### 3.5. RET Fusions

*RET* fusions are frequently detected in non-small cell lung cancer and thyroid cancer, but they have also been reported in CRC at a frequency of approximately 0.2% [66]. *RET* fusion-positive CRC is considered a distinct molecular subset characterized by a TMB-H and frequent MSI-H status [67]. These fusions result in constitutive activation of the *RET* receptor tyrosine kinase, triggering downstream pathways such as *MAPK*, *PI3K*/*AKT*, and *JAK*/*STAT*, which promote oncogenic signaling and tumor proliferation. The LIBRETTO-001 phase I/II trial evaluated selpercatinib in *RET* fusion-positive solid tumors, including ten CRC patients [68]. In this subgroup, the ORR was 20%, with a median PFS of 13.2 months and a median duration of response of 9.4 months. While the response rate was lower than that seen in NSCLC or thyroid cancer (about 60%) [69,70], it compared favorably to standard third-line options for CRC such as regorafenib or trifluridine/tipiracil. These findings highlight the potential utility of *RET* inhibition in this rare molecular subtype of CRC. In Japan, FoundationOne^®^ CDx is approved as a companion diagnostic test for *RET* fusions. The FDA approved selpercatinib and pralsetinib for treating *RET* fusion-positive solid tumors regardless of histology, whereas, in Japan, only selpercatinib is currently approved for *RET* fusion-positive solid tumors [68,71]. For CRC harboring *RET* fusions, selpercatinib is an effective treatment option.

### 3.6. NTRK Fusions

*NTRK* fusions involve rearrangements of *NTRK1*, *NTRK2*, or *NTRK3* with various upstream partners, leading to the constitutive activation of *TRK* signaling. This results in the continuous activation of downstream pathways such as *RAS*/*MAPK* and *PI3K*/*AKT*, contributing to oncogenesis across a wide range of solid tumors. The frequency of *NTRK* fusions in CRC is approximately 0.7% [72]. *NTRK* fusion-positive CRC is characterized by TMB-H, MSI-H status, and a mutually exclusive relationship with *RAS* and *BRAF* mutations. These features suggest that *NTRK* fusion-positive CRC may be particularly sensitive not only to *TRK* inhibitors but also to immune checkpoint inhibitors. In Japan, FoundationOne^®^ CDx and OncoGuide™ NCC Oncopanel System are approved for identifying *NTRK* fusions. Larotrectinib and entrectinib have received tumor-agnostic approval from the FDA and are also approved in Japan for solid tumors with *NTRK* fusions [73,74].

Clinical trials have shown remarkable efficacy of *TRK* inhibitors. Larotrectinib demonstrated an ORR of 75% across various tumor types, including CRC, with durable responses and minimal toxicity [73]. Similarly, entrectinib yielded an ORR of 57% and a median duration of response of 10 months in an integrated analysis of three phase 1–2 trials. Notably, entrectinib also showed intracranial activity due to its ability to cross the blood–brain barrier, an important consideration in patients with CNS involvement [74].

Given these findings, larotrectinib and entrectinib are now recommended as treatment options for *NTRK* fusion-positive CRC, especially in refractory cases lacking other actionable alterations. Continued routine testing for *NTRK* fusions in TMB-H and MSI-H CRC subtypes may help identify additional candidates for *TRK*-targeted therapies.

## 4. Emerging Diagnostic Technologies Not Yet Reimbursed in Japan

Several innovative technologies have emerged that hold great potential to transform the clinical management of colorectal cancer. These include circulating tumor DNA (ctDNA)-based monitoring for minimal residual disease (MRD), Neo-*RAS* conversion assessment via liquid biopsy, and DNA methylation profiling to predict therapeutic response. Although these technologies are clinically validated and already in use for research or pilot programs in Japan, they remain outside the scope of national insurance reimbursement, limiting their integration into routine clinical practice. Below, we describe each of these technologies in more detail and discuss their implications for precision medicine in CRC.

### 4.1. Circulating Tumor DNA and Molecular Residual Disease (MRD) Detection

ctDNA has emerged as a promising biomarker for detecting MRD and predicting recurrence risk in CRC. Unlike traditional tissue-based genetic testing, ctDNA analysis is a minimally invasive method that detects tumor-derived DNA fragments in the bloodstream, providing real-time insights into tumor dynamics. Recent studies have demonstrated the clinical utility of ctDNA in postoperative MRD monitoring, treatment response evaluation, and the early detection of relapse [75,76].

The CIRCULATE-Japan GALAXY observational study has shown that ctDNA-based MRD detection is strongly associated with recurrence risk and prognosis in CRC [77]. This study, which included over 2000 patients with stage II–III colon cancer and stage IV CRC, found that ctDNA positivity during the MRD detection window was significantly correlated with shorter DFS and OS. Patients with ctDNA positivity after surgery had an approximately 12-fold higher risk of recurrence and a nearly 10-fold higher risk of mortality compared with ctDNA-negative patients. Furthermore, patients who achieved ctDNA clearance through ACT experienced markedly improved DFS and OS, suggesting that ctDNA could serve as a predictive biomarker for treatment response.

Despite its clinical significance, ctDNA analysis is not yet covered by national insurance in Japan, limiting its widespread adoption in routine practice due to cost barriers. However, ongoing clinical trials are investigating its role in guiding adjuvant therapy decisions. The GALAXY study provided compelling evidence that ctDNA monitoring could stratify patients into high- and low-risk groups, thereby enabling more precise adjuvant treatment strategies. For instance, ctDNA-positive patients after surgery demonstrated a significant survival benefit from ACT, whereas ctDNA-negative patients showed no clear advantage from chemotherapy, suggesting that ctDNA analysis could help tailor treatment intensity [77].

However, despite its potential clinical utility, ctDNA analysis also presents several methodological limitations that must be considered before its widespread clinical adoption. Preanalytical factors such as blood collection tube type, processing time, and storage conditions can significantly affect cell-free DNA (cfDNA) yield and integrity [78]. For instance, delays in plasma separation may lead to leukocyte lysis and contamination with genomic DNA, which can reduce assay specificity.

Analytical sensitivity is another concern, especially in low-shedding tumors or early-stage settings where ctDNA levels may fall below the limit of detection. Assay performance varies by platform, with detection limits typically ranging from 0.01% to 0.1% variant allele frequency (VAF) [78]. Tumor fraction—the proportion of ctDNA within total cfDNA—is a key determinant of assay reliability, yet no consensus thresholds have been established for MRD detection.

Additionally, the clinical validity and utility of ctDNA results remain under investigation. Even with advanced tumor-informed assays such as Signatera™, false negatives can occur, and the correlation between ctDNA clearance and survival outcomes is still being studied [77,79].

The GALAXY study, an observational arm of the CIRCULATE-Japan project, represents one of the largest efforts to validate ctDNA-guided strategies in colorectal cancer [78,80]. While interim analyses have demonstrated strong prognostic value for MRD status, further standardization of assay procedures and harmonization across platforms are essential before ctDNA can be widely adopted in routine practice.

One major advantage of ctDNA analysis is its ability to detect residual disease earlier than conventional imaging techniques [75]. Longitudinal ctDNA monitoring has shown that ctDNA positivity often precedes radiological recurrence, potentially allowing for earlier intervention. In addition, ctDNA analysis enables the detection of specific genetic mutations, providing insights into acquired resistance mechanisms and facilitating treatment adjustments based on evolving tumor profiles.

Recent studies have investigated ctDNA analysis as a method for assessing the feasibility of reintroducing anti-EGFR therapy [39,40]. One clinically significant concept in this context is Neo-*RAS* wild-type CRC. This term refers to cases where a tumor that was initially *RAS*-mutant at diagnosis later becomes *RAS* wild-type during the course of treatment—a phenomenon observed through liquid biopsy using ctDNA analysis [38]. The OncoBEAM™ RAS CRC Kit, which utilizes BEAMing digital PCR technology, enables the highly sensitive detection of *RAS* mutations in ctDNA from plasma samples and has been applied in the assessment of Neo-*RAS* wild-type conversion [17]. In cases where this phenomenon is confirmed, reintroducing anti-EGFR therapy may provide meaningful clinical benefits [39,41]. However, the clinical utility of Beaming technology for detecting Neo-*RAS* wild-type conversion remains under investigation, and further studies are needed to establish its role alongside ctDNA-based treatment monitoring and MRD detection.

### 4.2. DNA Methylation

DNA methylation is an epigenetic modification that plays a crucial role in CRC development and progression, and its potential as a predictive biomarker for the efficacy of anti-EGFR antibody therapy has gained attention [33]. The methylation of CpG islands in the promoter regions of certain genes can lead to transcriptional silencing, thereby influencing tumor progression [81].

Recently, the OncoGuide™ EpiLight™ methylation detection kit, approved in Japan in June 2024, became the world’s first in vitro diagnostic test based on real-time PCR technology for assessing DNA methylation. This assay evaluates the methylation status of 16 specific genomic regions, reflecting genome-wide DNA methylation levels, and it serves as a decision support tool for selecting therapeutic agents in CRC. Tumor tissue DNA is subjected to bisulfite conversion and analyzed to classify cases as HMCC or low-methylated colorectal cancer (LMCC).

In a validation study involving 156 patients with metastatic CRC, LMCC patients with *RAS* wild-type tumors exhibited significantly higher response rates to anti-EGFR antibody therapy (33.3% vs. 4.2%, *p* = 0.004), longer PFS (6.6 vs. 2.5 months, HR = 0.22, *p* < 0.001), and improved OS (15.5 vs. 5.6 months, HR = 0.23, *p* < 0.001) compared to HMCC patients [82]. The predictive power of DNA methylation status remained significant in multivariate analyses, independent of primary tumor location and *RAS*/*BRAF* status, supporting its robustness.

These findings suggest that DNA methylation status serves as a predictive biomarker for anti-EGFR therapy efficacy in *RAS* wild-type CRC. Although the OncoGuide™ EpiLight™ methylation detection kit has received regulatory approval in Japan, it remains non-covered by insurance. Incorporating DNA methylation testing into precision medicine strategies is expected to enhance the selection of optimal treatment approaches for patients with CRC. Therefore, introducing insurance coverage is highly anticipated.

## 5. Implementation Challenges and Limitations

While recent advances in molecular diagnostics and targeted therapies have transformed the landscape of colorectal cancer care in Japan, significant limitations remain in the practical implementation of precision medicine. Despite nationwide initiatives and expanding insurance coverage, several structural and operational challenges restrict the full clinical integration of biomarker-guided strategies. Although Japan has established a robust infrastructure for precision medicine—including reimbursement for biomarker testing, national genomic initiatives such as c-CAT and SCRUM-Japan, and the designation of Cancer Genomic Medicine Hospitals—several practical and structural challenges continue to hinder its widespread clinical adoption.

One major limitation is the turnaround time for CGP, which often exceeds four weeks. This delay can impede timely treatment decisions, especially in patients with aggressive disease progression. Moreover, insurance coverage allows CGP testing only once per lifetime, limiting opportunities for re-evaluation in the face of clonal evolution or acquired resistance.

The low rate of genomically matched therapy administration—despite the detection of actionable alterations in approximately 40% of patients—is another critical concern. Barriers include restricted access to unapproved agents, the limited availability of clinical trials, and insufficient pathways for off-label drug use. While systems such as patient-requested therapy (PRT) and advanced medical care exist, their complexity and procedural burden constrain practical utilization.

Disparities in MTB implementation further complicate the precision medicine landscape. The structure and function of MTBs vary significantly across institutions, resulting in the inconsistent interpretation of genomic data and variable access to treatment recommendations. The standardization of MTB operations and expansion of remote or centralized consultation networks may be necessary.

In addition, promising technologies such as ctDNA for MRD detection, Neo-*RAS* conversion monitoring, and DNA methylation profiling remain non-reimbursed, restricting their integration into routine practice despite emerging clinical evidence.

Finally, there is a growing need to address gaps in genomic literacy among both healthcare providers and patients. Many patients lack a clear understanding of CGP results, especially when findings are inconclusive or require experimental treatment enrollment. Strengthening genetic counseling services and educational initiatives will be essential to support informed decision making.

These limitations underscore the importance of structural reform in Japan’s precision oncology ecosystem, including the expansion of insurance coverage, the decentralization of clinical trials, and the development of policies that promote timely, equitable, and inclusive and equitable access to molecularly guided therapies.

## 6. Ethical and Regulatory Considerations

The advancement of precision medicine in CRC has introduced a range of ethical and regulatory challenges, particularly in the context of genomic testing, data sharing, and access to targeted therapies. One of the primary ethical concerns involves informed consent for CGP. Unlike conventional tests, CGP may reveal incidental findings, germline mutations, or alterations with uncertain clinical significance. Ensuring that patients understand the scope, limitations, and possible psychosocial implications of such testing is critical. In Japan, pre-test genetic counseling is available, particularly at designated Cancer Genomic Medicine Hospitals, although it is not legally mandated. According to c-CAT, patients undergoing cancer genomic profiling have the option to receive genetic counseling prior to testing. Furthermore, in 2023, the Act on Promotion of Cancer Genomic Medicine was enacted to ensure ethical safeguards and prevent social disadvantage or discrimination, especially in cases where germline mutations are identified. Moreover, communicating uncertain or uninformative results can cause anxiety and confusion for patients, highlighting the need for more robust post-test counseling support. Data privacy and the secondary use of genomic information are also critical issues. Programs such as c-CAT collect and centralize genomic and clinical data from across the country, contributing to valuable real-world evidence. However, strict data governance frameworks are essential to protect patient confidentiality and prevent the misuse of sensitive genetic information. Japan’s Act on the Protection of Personal Information (APPI) governs such data use, but ongoing updates may be needed to address evolving concerns related to AI analysis and cross-border data sharing. From a regulatory standpoint, the approval and reimbursement of novel diagnostics and therapies remain complex. While CGP tests are now covered by national health insurance, access to off-label or investigational treatments based on CGP results is still restricted. Japan’s regulatory frameworks, including advanced medical care and patient-requested therapy systems, offer conditional pathways to access such drugs, but their complexity often limits practical implementation. There is a growing call for a more flexible mechanism akin to the U.S. FDA’s expanded access program or conditional approval schemes seen in the EU. Finally, equity in access represents a broader ethical issue. Geographic disparities in MTB capacity, variation in clinical trial availability, and socioeconomic barriers may limit the reach of precision oncology. Policies aimed at decentralizing MTBs and expanding telemedicine consultation platforms could help ensure that genomic-guided therapies are available to all eligible patients, regardless of location or income. As precision oncology becomes more integrated into routine care, addressing these ethical and regulatory challenges will be essential for ensuring responsible, equitable, and sustainable implementation.

## 7. Conclusions and Future Directions

Precision medicine strategies outlined in this review have contributed to improvements in treatment response, survival outcomes, and therapy selection efficiency, as supported by recent clinical trial data referenced in each biomarker-specific section. With advances in precision medicine for CRC, biomarker testing has become increasingly important in determining eligibility for molecular targeted therapies and immunotherapy. This review has outlined the current diagnostic and therapeutic strategies in Japan, focusing on *RAS*, *BRAF*, MSI/MMR, *HER2*, CGP, *RET* fusions, *NTRK* fusions, and ctDNA. Many of these biomarker tests are now covered by national insurance and play an essential role in guiding treatment selection. However, some promising technologies, such as ctDNA-based MRD detection and Neo-*RAS* wild-type conversion assessment, remain non-covered by insurance, highlighting the need for further evidence accumulation and policy development.

The use of CGP is expanding, as it enables the simultaneous detection of multiple genetic alterations and refines personalized treatment approaches. Liquid biopsy technologies, particularly ctDNA analysis, are also rapidly advancing and may revolutionize treatment monitoring, recurrence prediction, and early therapeutic intervention. The CIRCULATE-Japan GALAXY study has demonstrated the potential of ctDNA-guided decision making in adjuvant therapy, suggesting that future treatment paradigms will shift toward personalized adjuvant strategies based on MRD assessment rather than traditional clinicopathological factors alone. Further validation and integration of these approaches into clinical guidelines are crucial.

Emerging therapeutic developments, including *KRAS* G12C inhibitors, have expanded treatment options for previously untreatable patient subgroups. Ongoing research into *KRAS* G12D and *KRAS* G12V inhibitors, as well as pan-*KRAS* inhibitors, holds promise for overcoming resistance mechanisms and broadening therapeutic applicability. Similarly, *HER2*-targeted therapies, such as T-DXd, have demonstrated efficacy and may soon become a standard treatment option for *HER2*-positive CRC in Japan. These drugs have shown promising results, and their regulatory approval in Japan lags behind that in Western countries. Accelerating drug approval processes and expanding access to clinical trials will help bridge this gap.

Furthermore, real-world data from national initiatives, such as c-CAT and SCRUM-Japan, will play an essential role in optimizing biomarker-driven treatments. Integrating real-world data with clinical trial data can provide valuable insights into long-term treatment efficacy, safety, and cost-effectiveness. Future efforts should focus on leveraging real-world data to refine treatment algorithms, improve patient stratification, and identify novel therapeutic targets. The establishment of a nationwide precision medicine network that links real-world data with biomarker profiles could facilitate adaptive clinical trial designs, ultimately accelerating the development of new therapies and improving access to precision medicine in routine practice.

To ensure the effective integration of precision medicine into standard CRC care in Japan, several key priorities must be addressed. First, expanding insurance coverage for clinically validated technologies, such as ctDNA-based MRD detection and DNA methylation assays, is essential for equitable access. In parallel, the nationwide standardization and enhancement of MTBs will improve the consistency and quality of genomic data interpretation and therapeutic recommendations. Accelerating the domestic approval and reimbursement processes for emerging targeted therapies—particularly those already approved overseas—will further expand treatment options for patients with rare or refractory mutations. Additionally, expanding access to clinical trials through decentralized, basket, and adaptive designs can improve trial enrollment and early access to novel therapies.

While real-world data integration will complement these efforts, the most critical enabler will be sustained multidisciplinary collaboration among researchers, clinicians, regulators, and policymakers. Such coordination is essential to establish a sustainable precision medicine ecosystem and ensure that genomic innovations translate into equitable, high-quality care for all patients with colorectal cancer in Japan.

## Figures and Tables

**Table 1 ijms-26-05029-t001:** Approved next-generation sequencing panels in Japan.

	OncoGuide™ NCC Oncopanel System	FoundationOne^®^ CDx	GenMineTOP^®^	FoundationOne^®^ Liquid CDx	Guardant360^®^ CDx
Tissue/Liquid	Tissue/Blood	Tissue	Tissue/Blood	Blood	Blood
FDA Approval	×	○	×	○	○
PMDA Approval	○	○	○	○	○
Companion Diagnostics	○	○	×	○	○
Somatic Mutations	DNA 124	DNA 324	DNA 737, RNA 455	DNA 324	DNA 74
Fusion Genes	13	36	455	36	6
Germline Mutations (Secondary Findings)	○	△^※^	○	△^※^	△^※^
TMB	○	○	○	×	×
MSI	○	○	×	×	○

○: Applicable/Approved/Detectable. ×: Not applicable/Not approved/Not detectable. △^※^: Possible germline mutations; confirmation requires further testing. Abbreviations: FDA, Food and Drug Administration. PMDA, Pharmaceuticals and Medical Devices Agency. TMB, tumor mutational burden. MSI, microsatellite instability.

**Table 2 ijms-26-05029-t002:** Comparison of the TRIUMPH and MyPathway clinical trials evaluating HER2-targeted therapy in metastatic colorectal cancer.

	TRIUMPH Study	MyPathway Study
Design	Phase II, single-arm, Japan	Multicenter basket trial, U.S.
HER2 Definition	IHC 3+, FISH+, or Amplification determined by NGS (ctDNA)	IHC 3+ or FISH+
*RAS* Status	*RAS* wild-type	All *RAS* status
Line of Therapy	≥2nd line	≥2nd line
Treatment Regimen	Trastuzumab + Pertuzumab	Trastuzumab + Pertuzumab
ORR	30.0%	32.0%
Median PFS	4.0 months	2.9 months

Abbreviations: ORR, objective response rate. PFS, progression-free survival.

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
