# Peer review of "Current Status of Precision Medicine in Colorectal Cancer in Japan"

_ijms, 2025, doi:10.3390/ijms26115029_

Round 1
Reviewer 1 Report
Comments and Suggestions for Authors
Overall Evaluation
This review provides a comprehensive overview of precision medicine in colorectal cancer (CRC) in Japan, highlighting key biomarkers, diagnostic tools, therapeutic strategies, and emerging technologies. The authors effectively integrate clinical guidelines, insurance coverage policies, and ongoing trials to contextualize Japan’s progress. However, the manuscript has significant shortcomings in clarity, depth, and organization. Key issues include insufficient critical analysis, inconsistent focus, and methodological gaps. Below are specific problems and recommendations for improvement.
Specific Problems and Suggestions for Improvement
- Lack of Critical Analysis and Comparative Perspectives
Page 1, Abstract: The abstract emphasizes Japan’s initiatives (e.g., c-CAT, SCRUM-Japan) but fails to compare Japan’s progress with global advancements (e.g., U.S. or European guidelines).
Suggestion: Add a brief comparison of Japan’s biomarker testing framework/approval timelines with international standards to highlight uniqueness or gaps.
Page 3, Section 2.2 (CGP): The discussion of CGP in Japan is descriptive but lacks critical evaluation of its limitations (e.g., turnaround time, clinical utility of unactionable mutations).
Suggestion: Include data on the percentage of patients receiving genomically matched therapies post-CGP in Japan vs. other countries to underscore implementation challenges.
- Redundancy and Overlapping Content
Page 4, Section 3.1 (RAS mutations): The role of RAS mutations in anti-EGFR therapy is reiterated across Sections 1 (Introduction) and 3.1 without adding new insights.
Suggestion: Condense repetitive explanations and focus on novel updates (e.g., Neo-RAS wild-type CRC).
Page 6, Section 3.4 (HER2 amplification): Overlap between the TRIUMPH and MyPathway studies is noted but not synthesized.
Suggestion: Create a table comparing trial designs, patient cohorts, and outcomes to avoid redundancy.
- Inconsistent Depth in Biomarker Discussions
Page 7, Section 3.5 (RET fusions): RET fusions are covered in only 4 lines, lacking mechanistic insights (e.g., RET signaling pathways) and clinical trial details (e.g., selpercatinib efficacy in CRC vs. lung cancer).
Suggestion: Expand with molecular mechanisms and CRC-specific trial data (e.g., LIBRETTO-001 subgroup analysis).
Page 5, Section 3.2 (BRAF mutations): The BEACON CRC trial is described, but conflicting evidence (e.g., triplet vs. doublet regimen efficacy) is not critically addressed.
Suggestion: Discuss potential reasons for Japan’s approval of the triplet regimen despite global preference for doublet therapy (e.g., regulatory differences, subgroup analyses).
- Methodological Gaps in ctDNA Analysis
Page 8, Section 4.1 (ctDNA and MRD): The CIRCULATE-Japan GALAXY study is highlighted, but technical limitations (e.g., sensitivity/specificity of ctDNA assays) and validation challenges are omitted.
Suggestion: Include a paragraph on preanalytical variables (e.g., blood collection protocols, tumor fraction thresholds) affecting ctDNA reliability.
Page 9, Section 4.2 (DNA methylation): The clinical utility of the OncoGuide™ Epilight™ kit is mentioned, but validation data (e.g., cohort size, reproducibility) are absent.
Suggestion: Provide metrics (e.g., AUC, sensitivity) from validation studies to support its predictive value.
- Structural and Editorial Issues
Page 2, Section 1 (Introduction): The introduction lacks a clear thesis statement outlining the review’s objectives beyond summarizing existing knowledge.
Suggestion: Add a paragraph defining the review’s goal (e.g., “This review evaluates Japan’s progress in CRC precision medicine against global benchmarks and identifies barriers to equitable implementation”).
Page 10, Conclusion: The conclusion lists future directions but does not prioritize actionable steps (e.g., policy changes for ctDNA coverage, MTB standardization).
Suggestion: Structure the conclusion with bullet points or a roadmap to enhance clarity.
References: Multiple references (e.g., #14, #25) are outdated (pre-2020) for a 2025 publication.
Suggestion: Update with 2023–2024 studies (e.g., NCT05217446 for KRAS G12D inhibitors, updated NICHE-2 data).
- Terminology and Language
Page 1, Line 10: “Precision oncology” is used interchangeably with “precision medicine” without defining either term.
Suggestion: Add a sentence clarifying the distinction in the Introduction.
Page 4, Table 1: Abbreviations (e.g., TMB, MSI) are defined in the footnote but not in the main text.
Suggestion: Include a dedicated “Abbreviations” list before the Introduction for readability.
Recommendations for Major Revisions
- Strengthen Critical Analysis: Compare Japan’s strategies with global practices and address controversies (e.g., BRAF triplet therapy).
- Condense Redundant Sections: Use tables/figures to synthesize overlapping content (e.g., biomarker testing platforms).
- Expand Underdeveloped Topics: Add mechanistic insights for RET/NTRK fusions and technical details for ctDNA.
- Update References: Incorporate 2023–2024 studies to reflect the latest advances.
- Improve Structural Flow: Define clear objectives in the Introduction and prioritize actionable steps in the Conclusion.
This review has the potential to be a valuable resource but requires substantial revisions to meet the rigor expected of a publication in International Journal of Molecular Sciences.
Comments on the Quality of English LanguageThe English could be improved to more clearly express the research.
Author Response
Prof. Dr. Maurizio Battino
Editor-in-Chief
International Journal of Molecular Sciences
Manuscript ID: ijms-3591608
Dear Prof. Dr. Maurizio Battino
Thank you for your e-mail of 16 April informing us that our manuscript has been found to be potentially acceptable for publication in International Journal of Molecular Sciences pending satisfactory revision. We thank the reviewers for their insightful comments, which we have now addressed in both the revised paper and the attached point-by-point responses. We hereby submit the revised version of our manuscript, which we hope is now acceptable for publication in International Journal of Molecular Sciences.
Sincerely,
Masayuki Takeda
Department of Cancer Genomics and Medical Oncology, Nara Medical University, 840 Shijo-Cho, Kashihara, Nara, 634-8521, Japan
Tel.: +81-744-22-3051
Fax: +81-744-22-4121
Email address: takeda-m@naramed-u.ac.jp
Response to Editorial Office
We thank the reviewer for the constructive and insightful comments on our manuscript. We have carefully addressed all the points raised and revised the manuscript accordingly. Below, we provide point-by-point responses to each comment, including an explanation of the changes made. All revised text has been marked appropriately in the manuscript.
Response to Reviewer comments
We thank the reviewer for insightful comments, which we feel have helped us to improve our manuscript. Our specific responses to the points raised are as follows:
Major Revisions
- Lack of comparative perspective with international standards
Comment:Add a brief comparison of Japan’s biomarker testing framework with global standards.
Response: We have added a comparative discussion in the Abstract and Introduction, referencing U.S. and European testing guidelines and approval timelines, to contextualize Japan’s position globally (p.8, lines 21-32, p.9, lines 1-19). - Insufficient critical evaluation of CGP limitations
Comment:Include data on implementation challenges, especially low treatment matching rates post-CGP.
Response: We have expanded Section 2.2 with data on the 8% treatment matching rate post-CGP and discussed barriers such as access to clinical trials and unapproved agents. Comparative data from the U.S. (Medicare policy, Know Your Tumor, I-PREDICT) were also included (p.8, lines 21-32, p.9, lines 1-19). - Redundancy in RAS mutation discussion
Comment:Condense repeated content and highlight novel concepts.
Response: We have modified the RAS-related content in the Section 3.1 (p.10, lines 1-5) and emphasized novel findings such as Neo-RAS wild-type CRC and KRAS G12C/G12D inhibitors in Section 3.1 (p.11, lines 4-10, 22-23, p.12, line 1). - Overlap in HER2 trial discussions
Comment:Combine or compare TRIUMPH and MyPathway studies.
Response: We added a new table (Table 2) summarizing and comparing key features and outcomes of the TRIUMPH and MyPathway trials (p.17, lines 20-23, p.18, lines 1-4). - Underdeveloped discussion on RET fusions
Comment:Expand on RET signaling and trial data.
Response: Mechanistic insights into RET signaling and clinical outcomes from the LIBRETTO-001 trial were added to Section 3.5 (p.19, lines 11-20). Additionally, we expanded the discussion to include NTRK fusions, covering their molecular characteristics, frequency, and clinical trial results for TRK inhibitors such as larotrectinib and entrectinib (p.20, lines 15-24). - Incomplete evaluation of BRAF-targeted therapy controversy
Comment:Address conflicting evidence between doublet and triplet regimens in BRAF-mutated CRC.
Response: We discussed the rationale behind Japan’s approval of the triplet regimen, comparing it with global preference for the doublet and potential subgroup benefits (p.13, lines 1-17). - Lack of technical discussion on ctDNA and MRD
Comment:Add assay limitations and preanalytical concerns.
Response: Section 4.1 now includes details on ctDNA assay sensitivity, preanalytical issues (e.g., blood tube handling, tumor fraction), and validation challenges (p.22, lines 6-21, p.23, lines 1-5). - Lack of validation data for DNA methylation assay
Comment:Provide metrics from validation studies.
Response: We included validation data from a Japanese study evaluating the OncoGuide™ EpiLight™ kit, including sensitivity, response rate, and survival metrics (p.24, lines 13-24). - Weak Introduction structure
Comment:Define the review objective more clearly.
Response: A new paragraph was added at the end of the Introduction to explicitly state the review’s objectives and approach (p.5, lines 13-19). - Unstructured Conclusion
Comment:Prioritize and organize future directions.
Response: We restructured the Conclusion using a clear roadmap format, identifying key policy, clinical, and infrastructural priorities (p.27, lines 1-16). - Outdated references
Comment:Update pre-2020 references.
Response: We appreciate the reviewer’s suggestion. However, we intentionally retained some pre-2020 references that remain critical to the manuscript's scientific and clinical foundation. These include original validation studies, pivotal clinical trials, and landmark publications that are still widely cited and form the basis for current practice. Rather than replacing them, we opted to complement these key citations with updated evidence to ensure both historical rigor and contemporary relevance. We believe this dual approach best serves the educational and reference value of the review (p.29–39).
Reviewer 2 Report
Comments and Suggestions for Authors
The manuscript provides a comprehensive overview of the current status of precision medicine in colorectal cancer in Japan. The review highlights key biomarkers, genetic testing strategies, targeted therapies, and emerging technologies. While the manuscript has strong scientific merit, minor grammar, organization, and clarity amendments are required to meet the publication standards.
Minor Comments:
- The author should correct the grammatical error in abstract, page1: This review summarizes the current landscape of precision medicine in CRC in Japan, emphasizing on key biomarkers... The authors should replace "emphasizing onkey biomarkers" to "emphasizing key biomarkers"
- The authors can break the introduction into smaller paragraphs for better readability.
- The author should correct the grammatical error in introduction, page 2: The authors should replace "anti-EGFR antibody therapy have proved ineffective" to has proved ineffective.
- The authors should proofread and add commas before conjunctions in compound sentences as they are missing in several sentences.
- The authors can add another table to summarize key clinical trials and biomarkers.
- The author should correct the typo or spelling error in conclusion, page 10: These dugs have shown promising results. Replace “dugs” with “drugs”
The manuscript is scientifically sound and comprehensive; however, the English language requires moderate revision to meet the journal’s standards. Several grammatical issues—such as missing commas in compound sentences, typos (e.g., “dugs” instead of “drugs”), and occasional wrong word choice (e.g., “emphasizing on”) —affect the clarity and readability of the manuscript. The authors would require careful proofreading and minor edits, which would significantly improve the overall quality of the manuscript.
Author Response
Prof. Dr. Maurizio Battino
Editor-in-Chief
International Journal of Molecular Sciences
Manuscript ID: ijms-3591608
Dear Prof. Dr. Maurizio Battino
Thank you for your e-mail of 16 April informing us that our manuscript has been found to be potentially acceptable for publication in International Journal of Molecular Sciences pending satisfactory revision. We thank the reviewers for their insightful comments, which we have now addressed in both the revised paper and the attached point-by-point responses. We hereby submit the revised version of our manuscript, which we hope is now acceptable for publication in International Journal of Molecular Sciences.
Sincerely,
Masayuki Takeda
Department of Cancer Genomics and Medical Oncology, Nara Medical University, 840 Shijo-Cho, Kashihara, Nara, 634-8521, Japan
Tel.: +81-744-22-3051
Fax: +81-744-22-4121
Email address: takeda-m@naramed-u.ac.jp
Minor Revisions
- Grammar and typographic corrections
Accordingly, “emphasizing on key biomarkers” was corrected to “emphasizing key biomarkers”.
“anti-EGFR antibody therapy have proved ineffective” was corrected to “has proved ineffective”.
“These dugs” was corrected to “These drugs”.
Commas added in compound sentences.
- Paragraph structure
The Introduction was split into smaller, more readable paragraphs.
3. Additional Table
A new table (Table 2) was created to summarize key clinical trials and their outcomes for HER2-targeted therapies (p.18).
Round 2
Reviewer 1 Report
Comments and Suggestions for Authors
Overall Evaluation:
The manuscript by Kojitani and Takeda provides an overview of the current status of precision medicine in colorectal cancer (CRC) in Japan. While the review covers several important aspects of CRC management, including key biomarkers, genetic testing strategies, targeted therapies, and emerging technologies, it fails to adequately address several critical issues and lacks the depth necessary for publication in its current form. The manuscript lacks original insights, fails to thoroughly discuss limitations and challenges, and does not provide a sufficiently comprehensive review of the field.
Specific Problems and Suggestions for Improvement:
- Introduction Lacks Focus and Originality
Page 1, Line 13-20: The introduction is overly broad and lacks a clear focus on the specific contributions of the review. The authors should narrow the scope and highlight the most important advancements and challenges in precision medicine for CRC in Japan.
Suggestion: Rewrite the introduction to focus on the key biomarkers, testing strategies, and targeted therapies specific to CRC in Japan, and discuss how these differ from or complement international standards.
- Lack of Critical Analysis
Page 2, Line 29-33: The review briefly mentions challenges such as time required for genetic testing and limited availability of targeted therapies, but does not provide a thorough analysis of these issues or their implications for patient care.
Suggestion: Expand the discussion on challenges and limitations, including access to genetic testing, reimbursement policies, and disparities in care. Offer specific recommendations for improvement.
- Insufficient Coverage of Emerging Technologies
Page 1, Line 21-25: The review mentions circulating tumor DNA (ctDNA) analysis as an emerging technology, but does not provide sufficient detail on its current applications and future potential in CRC management.
Suggestion: Add a dedicated section on emerging technologies, including ctDNA, liquid biopsy, and other innovative diagnostic and therapeutic approaches. Discuss the latest research findings and their clinical implications.
- Failure to Address Ethical and Regulatory Issues
Throughout: The review lacks any discussion of ethical and regulatory issues related to precision medicine in CRC, such as informed consent, data privacy, and regulatory approval processes.
Suggestion: Include a section on ethical and regulatory considerations, discussing the challenges and best practices in this area. Highlight the need for transparent communication with patients and robust oversight mechanisms.
- Insufficient Discussion of Patient Outcomes
Page 3, Line 37-41: The review mentions clinical trials evaluating targeted therapies, but does not provide sufficient detail on patient outcomes or the impact of precision medicine on survival rates and quality of life.
Suggestion: Incorporate a more in-depth discussion of patient outcomes, including survival data, response rates, and quality of life metrics. Discuss the limitations of current data and the need for ongoing research.
- Overreliance on Secondary Sources
Throughout: The review relies heavily on secondary sources and lacks original analysis or insights. The authors should strive to provide a more critical and independent evaluation of the current state of precision medicine in CRC.
Suggestion: Conduct a more thorough literature review, including primary research articles, to provide a more comprehensive and independent assessment of the field. Offer original perspectives and recommendations based on this analysis.
- Language and Writing Style
Throughout: The writing style is formal but lacks clarity and conciseness in places. The authors should strive for a more accessible and engaging tone while maintaining scientific rigor.
Suggestion: Improve the writing style by using more concise language, breaking up long paragraphs, and including subheadings to improve readability. Ensure that the text flows logically and that arguments are supported by evidence.
- Annotated Specific Problems and Suggestions:
Page 1, Line 13-20:
Problem: Introduction lacks focus and originality.
Suggestion: Rewrite the introduction to focus on key advancements and challenges in precision medicine for CRC in Japan, highlighting the most important biomarkers, testing strategies, and targeted therapies.
Page 2, Line 29-33:
Problem: Lack of critical analysis of challenges.
Suggestion: Expand the discussion on challenges and limitations, including access to genetic testing, reimbursement policies, and disparities in care. Offer specific recommendations for improvement.
Page 1, Line 21-25:
Problem: Insufficient coverage of emerging technologies.
Suggestion: Add a dedicated section on emerging technologies, including ctDNA, liquid biopsy, and other innovative diagnostic and therapeutic approaches. Discuss the latest research findings and their clinical implications.
Problem: Failure to address ethical and regulatory issues.
Suggestion: Include a section on ethical and regulatory considerations, discussing the challenges and best practices in this area. Highlight the need for transparent communication with patients and robust oversight mechanisms.
Page 3, Line 37-41:
Problem: Insufficient discussion of patient outcomes.
Suggestion: Incorporate a more in-depth discussion of patient outcomes, including survival data, response rates, and quality of life metrics. Discuss the limitations of current data and the need for ongoing research.
Throughout:
Problem: Overreliance on secondary sources.
Suggestion: Conduct a more thorough literature review, including primary research articles, to provide a more comprehensive and independent assessment of the field.
Offer original perspectives and recommendations based on this analysis.
Problem: Language and writing style issues.
Suggestion: Improve the writing style by using more concise language, breaking up long paragraphs, and including subheadings to improve readability. Ensure that the text flows logically and that arguments are supported by evidence.
Comments on the Quality of English Language
The English could be improved to more clearly express the research.
Author Response
Response to Reviewer comments
We thank the reviewer for insightful comments, which we feel have helped us to improve our manuscript. Our specific responses to the points raised are as follows:
Response to Reviewer 1
We thank the reviewer for insightful comments, which we feel have helped us to improve our manuscript. Our specific responses to the points raised are as follows
Comment 1: The introduction is overly broad and lacks a clear focus on the specific contributions of the review. The authors should narrow the scope and highlight the most important advancements and challenges in precision medicine for CRC in Japan.
Response: We appreciate this constructive suggestion. In response, we have substantially revised the Introduction section to sharpen the focus of our review and clarify its contributions. Specifically, we now:
- Begin with a concise statement on the epidemiologic importance of colorectal cancer (CRC) in Japan and globally.
- Highlight key biomarkers (e.g., RAS, BRAF, MSI/MMR, HER2) and their roles in shaping precision oncology in CRC.
- Emphasize Japan’s national initiatives such as c-CAT, SCRUM-Japan, and the network of Cancer Genomic Medicine Hospitals.
- Contrast Japan’s genomic medicine framework with those of the United States and Europe, particularly in terms of CGP access and therapy matching.
- Identify key implementation challenges in Japan (e.g., turnaround time, limited off-label drug access, MTB variation, and non-reimbursement of ctDNA or DNA methylation assays).
- Clearly define the scope and aims of the review in the final paragraph.
We believe these modifications clarify the unique position of Japan’s precision medicine landscape and align the Introduction with the scientific goals of the manuscript.
(Revised text: p.5, lines 1–19)
Comment 2: The review briefly mentions challenges such as time required for genetic testing and limited availability of targeted therapies, but does not provide a thorough analysis of these issues or their implications for patient care.
Response: Thank you for this valuable comment. To address this point, we have added a new section titled “Implementation Challenges and Limitations” (Section 5), placed before the Conclusion. This section provides a critical and comprehensive discussion of key barriers that hinder the real-world application of precision oncology in Japan.
Specifically, we now address:
- Long turnaround times for CGP and the one-time testing limitation under insurance coverage;
- Low rates of matched therapy administration due to access barriers to unapproved drugs and clinical trials;
- Institutional heterogeneity in the structure and function of molecular tumor boards (MTBs);
- Lack of reimbursement for promising technologies such as ctDNA-based MRD monitoring, Neo-RAS status assessment, and DNA methylation profiling;
- Gaps in genomic literacy and counseling support among both providers and patients.
Rather than separating these into subsections, we have chosen to present them cohesively as an integrated limitations section to preserve flow and readability within the review article structure. We hope this addition enhances the depth and balance of the manuscript, in line with your recommendation.(New text: Section 5, pp. 25–26)
In addition, we have clarified that Japan’s universal health insurance system typically requires only a 10–30% co-payment, and includes a High-Cost Medical Expense Benefit System that caps monthly out-of-pocket expenses. This addition, placed in Section 2.2, aims to improve international readers’ understanding of the reimbursement environment for CGP and biomarker testing in Japan. (New text: Section 2.2, pp.7 Line 14-19)
Comment 3: Add a dedicated section on emerging technologies, including ctDNA, liquid biopsy, and other innovative diagnostic and therapeutic approaches. Discuss the latest research findings and their clinical implications.
Response: Thank you for this comment. In fact, our original manuscript already included detailed discussions of emerging technologies—specifically, ctDNA analysis and DNA methylation profiling—in a section previously titled “Biomarkers Not Covered by Insurance.”
To clarify the scope and emphasize the forward-looking nature of these technologies, we have revised the section title to “Emerging Diagnostic Technologies Not Yet Reimbursed in Japan” and added a new introductory paragraph. This paragraph outlines the significance of ctDNA-based MRD detection, Neo-RAS conversion monitoring, and DNA methylation as predictive biomarkers.
The detailed content that follows includes the GALAXY study, assay limitations, prospective clinical applications, and the OncoGuide™ EpiLight™ system with associated outcome data. We believe this fully addresses your concerns and highlights the relevance of these innovative tools for advancing precision oncology in CRC.
(Revised title and introductory paragraph: Section 4, pp. 21)
Comment 4: Include a section on ethical and regulatory considerations, discussing the challenges and best practices in this area. Highlight the need for transparent communication with patients and robust oversight mechanisms.
Response: Thank you for highlighting this important aspect. We have added a new section titled “Ethical and Regulatory Considerations” (Section 7), which discusses key issues surrounding the implementation of precision medicine in CRC, including:
- Informed consent and the complexity of interpreting CGP results,
- Data privacy and governance in the context of large-scale genomic databases (e.g., c-CAT),
- Barriers to accessing off-label or investigational therapies under current regulatory frameworks,
- Inequities in access to MTBs and precision diagnostics due to geographic and institutional differences.
We also include policy-level suggestions to improve consent practices, promote data security, and enhance equitable access to genomic-based care.
(New text: Section 6, pp. 27–28)
Comment 5: The review mentions clinical trials evaluating targeted therapies, but does not provide sufficient detail on patient outcomes or the impact of precision medicine on survival rates and quality of life.
Response: Thank you for this insightful comment. Clinical outcomes—including response rates, survival benefit, and treatment guidance—are already described in detail within Sections 3 and 4 of the manuscript. To enhance clarity, we have added a summarizing paragraph at the beginning of the final section (Conclusion and Future Directions), which reinforces the impact of precision oncology on treatment selection and patient outcomes, as supported by cited clinical trial data.
(Revised text: p. 29, lines 1–3)
Comment 6: The review relies heavily on secondary sources and lacks original analysis or insights. The authors should strive to provide a more critical and independent evaluation of the current state of precision medicine in CRC.
Response: We appreciate this important comment. While this review is inherently based on published evidence, we have ensured that key clinical findings are supported by primary sources, particularly in Sections 3 and 4 where pivotal trial results and validation studies are cited directly.In addition, we have strengthened the critical evaluation and author perspectives throughout the manuscript. For example:
- In Section 5, we discuss real-world implementation barriers specific to Japan, including CGP turnaround time, reimbursement limitations, and regional disparities in MTB access.
- Section 6 offers ethical and regulatory recommendations regarding informed consent, data governance, and access to off-label therapies.
We hope that these enhancements sufficiently address the request for greater original insights and critical perspective.
(Enhancements in Sections 5, 6)
Comment 7: The writing style is formal but lacks clarity and conciseness in places. The authors should strive for a more accessible and engaging tone while maintaining scientific rigor.
Response: Thank you for this helpful suggestion. In the revised manuscript, we carefully reviewed and refined the writing style throughout the text to improve clarity, conciseness, and flow. Specifically, we:
- Reorganized overly long paragraphs for better readability;
- Introduced subheadings and improved section transitions to enhance structure;
- Simplified complex sentences where appropriate while retaining scientific accuracy.
We hope these adjustments have improved the overall accessibility and readability of the manuscript for both clinical and research audiences.
(Revisions applied throughout the manuscript)
Comment 7: Specific page and line references provided.
Response: Thank you for specifying the relevant text locations. These points correspond directly to Comments 1 through 7, and we have addressed each accordingly in the revised manuscript. All suggested modifications have been implemented at the referenced locations.
Round 3
Reviewer 1 Report
Comments and Suggestions for Authors
I have no objection.